# The Bucherer–Bergs Multicomponent Synthesis of Hydantoins—Excellence in Simplicity

**DOI:** 10.3390/molecules26134024

**Published:** 2021-06-30

**Authors:** Martin Kalník, Peter Gabko, Maroš Bella, Miroslav Koóš

**Affiliations:** Institute of Chemistry, Center for Glycomics, Slovak Academy of Sciences, Dúbravská cesta 9, SK-845 38 Bratislava, Slovakia; martin.kalnik@savba.sk (M.K.); chempega@savba.sk (P.G.); maros.bella@savba.sk (M.B.)

**Keywords:** hydantoins, aldehyde, ketone, multicomponent reaction, Bucherer–Bergs reaction

## Abstract

Hydantoins and their hybrids with other molecules represent a very important group of heterocycles because they exhibit diverse biological and pharmacological activities in medicinal and agrochemical applications. They also serve as key precursors in the chemical or enzymatic synthesis of significant nonnatural α-amino acids and their conjugates with medical potential. This review provides a comprehensive treatment of the synthesis of hydantoins via the Bucherer–Bergs reaction including the Hoyer modification but limited to free carbonyl compounds or carbonyl compounds protected as acetals (ketals) and cyanohydrins used as starting reaction components. In this respect, the Bucherer–Bergs reaction provides an efficient and simple method in the synthesis of important natural products as well as for the preparation of new organic compounds applicable as potential therapeutics. The scope and limitations, as well as a comparison with some other methods for preparing hydantoins, are also discussed.

## 1. Introduction

The Bucherer–Bergs reaction is one of the most convenient general methods for the preparation of 5-substituted and 5,5-disubstituted hydantoins (imidazolidine-2,4-diones, 2,4-dioxoimidazolidines). Although the reaction was first discovered by Bergs [1] (but the first formation of 5,5-dimethylhydantoin from a mixture of acetone and hydrocyanic acid exposed to sunlight for a period of 5–7 months was observed by Ciamician and Silber in 1905 [2]), it is usually credited to Bucherer, who elaborated most of the experimental conditions and applications [3,4,5]. Generally, in this multicomponent reaction, the aldehyde or ketone in aqueous ethanol is heated at 60–70° with potassium (or sodium) cyanide and ammonium carbonate to produce directly hydantoins **1** (Scheme 1).

This reaction works well for aliphatic and aromatic aldehydes or ketones and for cyclic ketones despite some reports concerning the failure of this reaction. For such difficult cases, the use of acetamide (formamide as well as dimethylformamide) as a solvent has been recommended [6,7]. It was found that ultrasonication could also accelerate hydantoin formation [8]. Alternatively, better yields of hydantoins offer the Hoyer modification [9]. In this case, the standard reaction mixture is heated in the atmosphere of CO_2_ in a closed system at elevated pressure. Because of the wide applicability of the Bucherer–Bergs reaction, it has formerly been proposed as an analytical method for identifying ketones [10].

Hydantoins may be regarded as cyclodehydrated hydantoic acids (α-ureido acids), and this is reflected in their properties because both these compounds are readily interconvertible. Several natural or synthetic hydantoins themselves or their conjugates with other molecules exhibit diverse biological and pharmacological activities in medicinal, such as antimicrobial [11,12,13,14,15], antiviral [16,17,18], antitumor [19,20,21,22], antiarrhythmic [23,24,25,26], anticonvulsant [27,28,29,30,31,32,33,34], antihypertensive [35], antidiabetic [36,37,38,39], and agrochemical, such as herbicidal and fungicidal [40,41,42,43,44,45], applications. The studies on the biological activities of hydantoins has made great progress during the last three decades, and hydantoin derivatives have been therapeutically applied or are in the stage of investigation (Figure 1). For example, Phenytoin (Phenytek^®^, Dilantin^®^, Epanutin^®^, Diphenin^®^)—an antiepileptic drug—is still the drug of choice for the treatment of generalized tonic–clonic seizures (grand mal epilepsy) and focal motor seizures [29,46,47,48,49]; today, Phenytoin has found new applications because of the neuro- and cardioprotective properties [50,51]; Mephenytoin (Mesantoin^®^; it is no longer available in the US or the UK) and Fosphenytoin (Cerebyx^®^, Prodilantin^®^) are also effective anticonvulsants, the latter is used only in hospitals for the short-term (five days or less) treatment of epilepsy [52]; Nitrofurantoin (Furadantin^®^, Macrobid^®^, Macrodantin^®^) and Nifurtoinol (Urfadyn^®^)—produces antibacterial activity effective for the treatment of urinary tract infections [53,54,55]; Nilutamide—produces an antiandrogenic effect in the treatment of an advanced stage of the carcinoma of the prostate [19,20,22]; Sorbinil—an aldose reductase inhibitor that blocks the formation of sorbitol from excess glucose and thus may prevent many diabetic neuropathies [56,57,58]; Dantrolene (Dantrium^®^)—used to treat malignant hyperthermia, neuroleptic malignant syndrome, ecstasy intoxication, and muscle spasticity (stiffness and spasms) caused by conditions such as a spinal cord injury, stroke, cerebral palsy, or multiple sclerosis and is currently the only specific and effective treatment for malignant hyperthermia [59]; Azimilide—an investigational class III anti-arrhythmic drug that blocks fast and slow components of the delayed rectifier cardiac potassium channels (until now, it has not been approved for use in any country but is currently in clinical trials in the United States) [60]. Iprodione (Rovral^®^, Kidan, Glycophene) is an example of a commercially used fungicide [61]. Because of their unique features, some glycofuranosylidene- and glycopyranosylidene-spiro-hydantoins have received wide attention. For example, (+)-hydantocidin (D-ribofuranosylidene-spiro-hydantoin) [62,63] possesses significant herbicidal and plant growth regulatory activities [41,64,65,66]; glucopyranosylidene-spiro-hydantoin [36,67,68] is among the most potent inhibitors of rabbit muscle glycogen phosphorylase known to date (*K*i = 3–4 µM).

Additionally, hydantoins also serve as key precursors in the chemical or enzymatic synthesis of significant nonnatural α-amino acids and their conjugates with medical potential. In this respect, the Bucherer–Bergs reaction provides an efficient method in the synthesis of important natural products as well as for the preparation of new organic compounds applicable as potential therapeutics.

Until now, five relevant reviews [69,70,71,72,73] and one book chapter [74] have appeared regarding the chemistry of hydantoins covering, inter alia, some aspects of the Bucherer–Bergs reaction. This review provides a comprehensive treatment of the synthesis of hydantoins via the Bucherer–Bergs reaction including the Hoyer modification but limited to free carbonyl compounds or carbonyl compounds protected as acetals (ketals) and cyanohydrins used as starting reaction components (i.e., the “classical” Bucherer–Bergs reaction starting from carbonyl compounds). The synthesis of hydantoins starting from corresponding amino nitriles (prepared from carbonyl compounds in a separate reaction step) or imines (prepared separately from carbonyl compounds or cyanides) were not included because, in this synthetic modification, only two reaction components are comprised, so these reactions are not multicomponent. Analogously, the other synthetic methods affording hydantoins were not reviewed in this review.

## 2. Mechanism and Stereochemistry

Since the action of ammonium carbonate on cyanohydrins **2** and α-amino nitriles **3** under identical reaction conditions also yields hydantoins, Bucherer himself proposed [3] that they are probably the first intermediates of this reaction. The last intermediates, prior to ring closure, may be either an N-substituted carbamic acid **4** or a corresponding carbamide **5**, although this has not been established experimentally. The last step would then involve either the formation of a 5-iminooxazolidin-2-one ring **6** affording hydantoin via isocyanate intermediate **7** (Scheme 2) or the addition of an amino group to the nitrile in carbamide **5** to closure of the 4-imino-2-oxoimidazolidine ring **8** followed by hydrolysis to the corresponding hydantoin (Scheme 3).

Treatment of α-amino nitriles with carbon dioxide also provided the disubstituted ureas **9**, which underwent cyclization in water at room temperature followed by hydrolysis of the imine **10** to the corresponding 3-N-substituted hydantoin **11** (Scheme 4) [75,76]. However, α-amino nitriles **2** are generally accepted as intermediates in the Bucherer–Bergs synthesis producing 1,3-unsubstituted hydantoins **1** instead of products like **11**. The participation of intermediary α-amino nitrile is supported by the fact that carbon disulphide also ring-closes such compounds to corresponding 2,4-dithiohydantoins [77,78].

According to the general stereochemical outcome of the Bucherer–Bergs reaction [71], the thermodynamically controlled spiro products are obtained with the C-4 carbonyl group of the imidazolidine-2,4-dione ring in the less hindered position. Thus, Munday [79] found that the Bucherer–Bergs reaction of 4-*tert*-butylcyclohexanone (**12**) (Scheme 5) predominantly afforded one isomeric hydantoin **13** (designated α) and only a trace of a second isomer **14** (designated β).

Although Cremlyn and Chisholm [80] reversed this assignment, it was later established [81] by unequivocal chemical evidence that the major isomer of two isomeric 4-benzoyloxycyclohexane-1-spiro-5′-hydantoins (**16** and **17**) (Scheme 5) obtained by the Bucherer–Bergs reaction from 4-benzoyloxycyclohexanone (**15**) had the structure of **16** (designated α) thus supporting, by analogy, Munday’s assignment. More direct evidence for this assignment came from ^13^C-NMR and UV spectra as well as from acetylation of α- and β-hydantoins **13** and **14** [82]. Additionally, mechanistic considerations of the Strecker and Bucherer–Bergs reactions enabled an explanation of how the same amino nitrile can yield either the α- or the β-hydantoin, according to the reaction conditions. On mechanistic grounds, it seems reasonable that, during the Strecker reaction, various equilibria are established rapidly in alkaline solution (via the intermediacy of **19** and **20**) (Scheme 6) but not in acidic solution.

Even in the weakly alkaline solution produced by dissolving the crude amino nitrile in aqueous ethanol, the rates of interconversion **21** ↔ **22** (Scheme 6) are very fast. It was confirmed [80,82] that the amino nitrile reacts with cyanic acid in acetic acid to form a urea derivative **23**, which can be cyclized to the β-hydantoin **14** via possible intermediates **24** and **25** (Scheme 7).

The important fact is that under acidic reaction conditions the interconversion **21** ↔ **22** does not take place and the β-isomer **14** is the main reaction product. However, if the same amino nitrile is treated with carbon dioxide in aqueous ethanol, the α-hydantoin **13** is obtained because the interconversion **21** ↔ **22** is rapid under these conditions. The possible mechanism of its formation (Scheme 8) is analogous with the general mechanism formerly proposed by Bucherer [3].

The preferential formation of the α-hydantoin indicates that the reaction route from reactants to the rate-determining step leading to its formation involves a lower overall energy barrier than does the route for hydantoin of β-series. It seems very likely that the rate-determining step on the path to **14** is **27** ↔ **29**. According to the Hammond principle, if this step is endothermic, the transition state will resemble **29**, which is, however, subject to considerable steric hindrance because of the compression between the 3,5-axial hydrogen atoms and the formation of C=NH group. Consequently, this path is a highly disfavored one. On the other hand, the path leading to the α-hydantoin (**22** → **21** → **26** → **28** → **30** → **13**) is less favored in its earlier, pre-equilibrium steps. Particularly, because of steric reasons, the conversion **21** → **26** is less favored than **22** → **27**. However, the relative rates by both the α- and β-paths depend upon the overall energy barrier between **22** and the intermediate after the rate-determining step (Hammett–Curtin principle), and this is, for steric reasons, larger on the β-path and, therefore, despite the abovementioned disfavoring, the α-hydantoin is formed preferentially.

## 3. Scope and Limitations

The scope of the Bucherer–Bergs synthesis is such that all reaction components including organic aldehydes and ketones are readily accessible, thus providing entry into a wide variety of 5-substituted and 5,5-disubstituted hydantoins. In addition, most of the final hydantoins are crystalline products and their isolation and purification is very simple. In most cases, one crystallization from a suitable solvent affords pure products. Despite the relative ease of execution and good yields, which make the Bucherer–Bergs reaction one of the most practical and suitable route to prepare hydantoins, several disadvantages and limitations to its applicability were found. One of the limitations is that it only has one point of diversity. Only changes in the structure of the starting ketone can affect variations of the final hydantoin.

In principle, the aldehyde or ketone parts R and R^1^ (Scheme 1) may be represented by a hydrogen atom, an alkyl, or a cycloalkyl, as well as an aryl, group. Generally, ketones are more suitable substrates to afford hydantoins unambiguously. However, the presence of functionality in the R and R^1^ of the starting aldehyde or ketone can complicate the formation of desirable products dramatically. Although the reaction is tolerant of a diverse array of functional groups, because of a strong basicity of the reaction mixture, the Bucherer–Bergs reaction is intolerant of alkali labile functional groups that may be present on the starting carbonyl substrate. Depending on their character, this intolerance may lead to simple deprotection (like deacylation if acylated hydroxyl groups are present), restoring unprotected functionality, or the present functionality may be changed to a new group (e.g., hydrolysis of nitrile, ester, amide, etc.) or to a reactive intermediate (e.g., carbanions in the case of nitroalkyl functionality). Moreover, the aqueous reaction conditions limit the application of starting ketones or aldehydes only to those which are stable under these conditions.

Similarly, the presence of powerful nucleophiles (amino and cyano groups) in the reaction mixture excludes the presence of readily substituted functionalities (like triflate, tosylate, or mesylate and halogen atoms) in the R and R^1^ of the starting aldehyde or ketone unless especially cyano- or amino-substituted final derivatives are desirable.

An unusual obstruction to the preparation of hydantoins is seen when an unprotected hydroxyl group is present in the α-position of the starting ketone (Scheme 1, R = not H, R^1^ = CH_2_OH). It was found that, in such cases, starting from sugar ketone **32**, the corresponding 4-carbamoyl-2-oxazolidinone **33** is formed preferentially instead of the expected hydantoin (Scheme 9) [83]. To obtain hydantoin products, appropriate protection of hydroxyl group (e.g., tritylation) prior to the Bucherer–Bergs reaction is necessary.

The anomalous Bucherer–Bergs reaction was observed when some carbohydrates with the free aldehyde group and *O*-isopropylidenated in the α-position were used as a starting material [84]. In these cases, the expected hydantoins were not formed, but a mixture of unsaturated hydantoin derivatives with the *Z* configuration and 5,5-dimethylhydantoin were obtained indicating that the 1,3-dioxolane ring (acetal group) vicinal to the aldehyde group is opened via an elimination reaction under formation of a double bond and that the liberated acetone undergoes the normal Bucherer–Bergs reaction to afford 5,5-dimethylhydantoin. Although the proportions of 5,5-dimethylhydantoin and unsaturated hydantoins formed are similar, as isolation of latter compounds is difficult, 5,5-dimethylhydantoin is always isolated as a major product, and the yields of unsaturated hydantoin derivatives depend very much on the structure of the starting material. Scheme 10 is illustrative for starting 2,3:4,5-di-*O*-isopropylidene-D-arabinose (**34**) and 2,3:4,5-di-*O*-isopropylidene-D-ribose (**35**). Because the chirality of C-2 is destroyed during the elimination reaction, the same products—5-(D-*erythro*-2-hydroxy-3,4-isopropylidenedioxybutylidene)imidazolidine-2,4-dione (**36**) and 5,5-dimethylhydantoin (**37**) resulted from both the starting D-*arabino* and D-*ribo* isomers **34** and **35**.

Similarly, 2,3:5,6-di-*O*-isopropylidene-D-xylose, 2,3:5,6-di-*O*-isopropylidene-α-D-mannofuranose (**38**), 1,2:3,4-di-*O*-isopropylidene-α-D-*galacto*-hexodialdo-1,5-pyranose (**39**), 2,3:4,5-di-*O*-isopropylidene-β-D-*arabino*-hexosulo-2,6-pyranose (**40**), and 1,2-*O*-isopropylidene-3-*O*-methyl-α-D-*xylo*-pentodialdo-1,4-furanose (**41**) also undergo anomalous reactions. Starting from **38**, a mixture of D-*glycero-*D*-galacto*- and D-*talo*-heptonic acid δ-lactone derivatives **42** (isolated in the form of acetates **43**) was obtained as a major product, together with a minority of unsaturated derivative **44** and 5,5-dimethylhydantoin (**37**) (Scheme 11).

Compound **39** afforded, in addition to the major product 5,5-dimethylhydantoin (**37**), the hydantoin derivative **45**, which is the product of the normal reaction, the diastereomeric 6-ureidohepturonamide **46**, and instead of the expected unsaturated hydantoin derivative, only a very low yield of saturated compound **47** (Scheme 12).

Compound **40** provided a mixture of diastereomeric cyanohydrins **48** and hydroxyamides **49**, together with 5,5-dimethylhydantoin (**37**) (Scheme 13). The high yield (78%) of **37** indicates that **40** is converted, via the anomalous reaction, mainly into unsaturated hydantoin derivative which, however, is unstable and decomposes.

Compound **41**, which contains an aldehyde group in the α position to an acetal-linked oxygen of an oxolane and not a dioxolane ring, yielded diastereomeric ureidohexuronamides **50** and **51** (the side-products of the normal reaction), 5,5-dimethylhydantoin (**37**), and the pyrido-imidazole derivative **52** (isolated as diacetate **53**), which can be formed from the acyclic intermediate **54** arising from the anomalous reaction (Scheme 14).

A similar anomalous reaction was also observed with starting methyl 2,3-*O*-isopropylidene-α-D-*lyxo*-pentodialdo-1,4-furanoside (**55**) [85]. However, because the elimination step involves a methoxy group at C-1, and not 1,2-*O*-isopropylidene-group-liberating acetone, thus excluding the formation of 5,5-dimethylhydantoin, corresponding cyanohydrin **56**, uronamide **57**, uronic acid **58**, and the pyrido[2,1-*e*]imidazolidine derivative **59** were isolated as main products in this case, together with a minority of ureidouronamide **60** (Scheme 15). Analogously to the formation of pyrido-imidazole derivative **52** from **54** (see Scheme 14), the pyrido[2,1-*e*]imidazolidine derivative **59** can be formed via intramolecular cyclization of the precursory aldehyde–unsaturated hydantoin derivative **61**.

The presence of a group at the anomeric C-1 position capable of elimination (like methoxyl in **55**) seems to be crucial for the anomalous course of the Bucherer–Bergs reaction (formation of the pyrido-imidazolidine products). This is because when 6-*O*-(*t*-butyldiphenylsilyl)-3,4-*O*-isopropylidene-2,5-anhydro-D-allose (**62**) was subjected to the Bucherer–Bergs reaction, only hydantoin **63** (i.e., the product of normal Bucherer–Bergs reaction) was isolated in 79% yield (Scheme 16), which after deprotection afforded (±)-5-(β-D-ribofuranosyl)-hydantoin (**64**) [86] a close analogue of naturally occurring biologically active Showdomycin. Contrary to **55** having a methoxyl group at C-1 and a formyl group at C-4 positions of the furanose ring, the C-4 position in **62** is occupied by a protected hydroxymethyl group, and the formyl group is positioned at the C-1 atom (regarding the compound name and atom numbering according to carbohydrate nomenclature, in the case of compound **62**, the C-4 and C-1 positions of furanose ring are, in fact, the C-5 and C-2 positions).

Although low stereoselectivity for simple carbonyl substrate is a general drawback of the Bucherer–Bergs reaction, both the rate and enantioselectivity of this reaction can be influenced by steric and electronic effects of substituents on the substrate. The suitable choice of substitution can lead to the predominancy of one enantiomer. On the other hand, electronic conditions and steric hindrance (due to the presence of the bulky substituents R and R^1^ as well as their unfavorable steric orientation) in starting ketone or aldehyde can even prevent successful formation of hydantoins. Thus, the resistance of 1,2:4,5-di-*O*-isopropylidene-β-D-*erythro*-2,3-hexodiulo-2,6-pyranose to the Bucherer–Bergs reaction may be explained, besides by unfavorable steric conditions, in terms of the interactions between the permanent dipoles about the anomeric group with those formed during the development of the transition state.

Because of the presence of organic and inorganic reaction components in the reaction mixture, the choice of the solvent is limited to very polar hydroxylic solvents like water, ethanol, and methanol. Most commonly, a mixture of one of these alcohols with water is used. It was established [87] that THF is also tolerated in the Bucherer–Bergs reaction but only at low concentrations in solvent mixtures with water and ethanol. The experiments performed on *n*-butyl phenyl ketone (1 mmol scale) under a standard set of reaction conditions (three equiv. of KCN, six equiv. of (NH_4_)_2_CO_3_, 75 °C, 24 h) and varying the reaction solvent (total volume constant at 9 mL) have shown no appreciable conversion (<15%) to the corresponding hydantoin in a binary solvent system THF–H_2_O (1:1). Similar results were obtained using the ternary solvent system THF–H_2_O–EtOH (2:1:1). However, reducing the amount of THF (THF–H_2_O–EtOH, 1:4:4) did improve the conversion to 47%. Complete conversion (>95%) enabling isolation of corresponding hydantoin in a 77% yield was achieved using these later reaction conditions when a sealed tube was used to prevent the release of the ammonia and carbon dioxide generated.

Among the disadvantages of the Bucherer–Bergs reaction, it has to be mentioned that the reaction component KCN (or NaCN) is classified as very toxic and dangerous for the organisms and environment and, therefore, the experiments must be performed very carefully by qualified individuals using appropriate protective equipment and respecting all risk and safety precautions for working with such highly hazardous material. Moreover, because of released toxic ammonia, the reactions should only be carried out in a fume cupboard.

## 4. Application to Synthesis

### 4.1. Overview

The primary significance of the Bucherer–Bergs reaction lies in the preparation and the many uses of the hydantoin products. Foremost among these uses is the ready access to starting carbonyl compounds and their enormous structural diversity. The possible transformation of hydantoins to the variety of α-amino acids under basic or acidic conditions represents another significant synthetic utility and potential of the Bucherer–Bergs reaction. Thus, a scalable process to prepare the INOS inhibitor PHA-399733, as a potential candidate for the treatment of osteoarthritis, asthma, and neuropathic pain was reported (Scheme 17) [88], using the Bucherer–Bergs hydantoin synthesis as the key step to introduce the amino acid group in the final molecule.

The Bucherer–Bergs reaction was also employed to prepare a key intermediate hydantoin for the synthesis of methionine amide (LY2140023), the first drug (a clinical candidate) acting on mGlu receptors that has been studied in humans to treat schizophrenia (Scheme 18) [89].

Transformation of hydantoins to α-amino acids proceeds through the intermediacy of ureido acids or ureido amides, which, in many cases, can be isolated as useful (new synthetic blocks, potential biological activity, etc.) individual compounds. Moreover, in some cases, ureido acids or ureido amides may result even as the main products of the Bucherer–Bergs reaction directly.

Furthermore, the hydantoins are accessible to further modifications applying e.g., N-alkylation; the Horner–Wadsworth–Emmons reaction; and aldol-type, cycloaddition and complexation reactions, thus affording additional synthetic routes to interesting new compounds. In addition, they are important heterocyclic scaffolds that induce biological effects, and they have pharmacological importance (see Section 1).

### 4.2. Applications in the Synthesis of Natural Products and Biologically Active Compounds

Hydantoin is an important heterocyclic core that exists in many naturally occurring products, mostly of marine organisms but also of bacteria. Most of them represent rather complicated structures with an incorporated hydantoin core. In many cases, the Bucherer–Bergs reaction in particular has been applied for the synthesis of this core, thus providing intermediary starting hydantoins necessary for further structural modification affording final biologically active compounds. Several compounds with a hydantoin structural unit in their molecules have been therapeutically applied, especially during the last three decades (see Section 1), and the Bucherer–Bergs method has been a choice for their preparation. The following biologically active hydantoins synthesized using the Bucherer–Bergs reaction could be mentioned:

#### 4.2.1. Sorbinil

This spirohydantoin aldose reductase inhibitor (for treatment of diabetic neuropathy), which is, according to IUPAC, (4*S*)-6-fluoro-2,3-dihydrospiro[4*H*-1-benzopyran-4,4′-imidazolidine]-2′,5′-dione, was first reported by Sarges in 1978 [90]. It was originally prepared by a multi-step process that essentially involved condensing 6-fluoro-4-chromanone with potassium cyanide and ammonium carbonate in ethanol under standard Bucherer–Bergs conditions to provide the corresponding racemic precursor of sorbinil (Scheme 19), followed by resolution of the latter (±)-compound with (–)-brucine to isolate the pharmacologically active *S*-(+)-enantiomer.

Sorbinil is obtained in a novel manner by optical resolution of racemic 2,3-dihydrospiro-6-fluoro[4*H*-l-benzopyran-4,4′-imidazolidine]-2′,5′-dione either (a) by direct resolution via the (–)-3-aminomethylpinane salt of sorbinil or (b) by a double resolving agent technique via a mother liquor concentrate of either the (+)-3-amino-methylpinane or the (–)-2-amino-2-norpinane salt of sorbinil, followed by the quinine salt of sorbinil [91].

The Bucherer–Bergs synthesis was also used for the preparation of ^14^C-labelled sorbinil, starting from 2,3-dihydro-6-fluoro-4*H*-1-benzopyran-4-one and ^14^C-potassium cyanide, followed by brucine resolution of the racemic spirohydantoin [92]. Tritiated sorbinil was obtained in two steps: (1) preparation of 8-chloro-sorbinil using the Bucherer–Bergs synthesis; (2) reductive dehalogenation of this 8-chloro substituted analog using tritium gas in the presence of triethylamine [93].

The Bucherer–Bergs reaction was also applied for the preparation of 2-methylsorbinil, i.e., (4S)(2*R*)-6-fluoro-2-methyl-spiro-[chroman-4,4′-imidazolidine]-2′,5′-dione. In this case, 6-fluoro-2-methyl-4-chromanone was condensed with potassium cyanide and ammonium carbonate in the usual manner to ultimately afford (±)-6-fluoro-2-methyl-spiro-[chroman-4,4′-imidazolidine]-2′,5′-dione in the form of the desired diastereoisomer. Resolution of the latter racemic compound with an aqueous quinine methohydroxide solution then finally gave the desired (4*S*)(2*R*)-isomer [94].

Analogously, the Bucherer–Bergs reaction with/without subsequent resolution of racemic spirohydantoins was used [93,94,95,96] for the preparation of many other sorbinil-like structural analogs of general formula (Figure 2).

#### 4.2.2. Phenytoin

This commonly used antiepileptic diphenylhydantoin (IUPAC name: 5,5-diphenylimidazolidine-2,4-dione) was first synthesized from hydroxy-diphenyl-acetic acid and urea by Biltz in 1908 [97]. Starting from benzophenone, under standard reaction conditions of the Bucherer–Bergs synthesis [(NH_4_)_2_CO_3_, NaCN, 60% EtOH, 58–62 °C, 10 h], phenytoin was obtained only in a 7% yield. Prolongation of the reaction time (90 h) increased the yields to 67%. Improved yields (75%) were obtained when the reaction mixture was heated at 110 °C in a closed vessel to retain the volatile components. Finally, the highest yields (91–96%) resulted using KCN instead of NaCN and propylene glycol or melted acetamide as a solvent in a steel bomb (Scheme 20) [98].

#### 4.2.3. Aplysinopsins

As to the chemical structure, naturally occuring aplysinopsins are, in general, 5-[heteroarylmethylidene]substituted hydantoins, more specifically, derivatives of 5-[(1*H*-indol-3-yl)methylidene]imidazolidine-2,4-dione or 5-[(1*H*-indol-3-yl)methylidene]-2-iminoimidazolidine-4-one (Figure 3), which can be isolated from various marine organisms (sponges, corals, etc.) [99,100,101,102,103].

They have aroused considerable interest especially because of their specific cytotoxicity for cancer cells [102] and their ability to affect neurotransmitters [103]. Among several synthetic approaches towards aplysinopsin-type structures, the Bucherer–Bergs reaction has been applied for the preparation of starting the hydantoin core. Thus, in a three-step synthesis of some aplysinopsins, the basic hydantoin prepared in the first step [104] by the Bucherer–Bergs reaction is transformed, in the next step, into (*Z*)-5-[(dimethylamino)methylidene]imidazolidine-2,4-dione or (*Z*)-5-[(dimethylamino)methylidene]-3-methylimidazolidine-2,4-dione using (*tert*-butoxy)bis(dimethylamino)methane (Bredereck’s reagent) or *N*,*N*-dimethylformamide dimethyl acetal (DMFDMA), respectivelly. These hydantoins react in the third step with indole to provide aplysinopsin derivatives (Scheme 21) [105].

#### 4.2.4. Hydantocidin

This spironucleoside metabolite isolated from the fermentation broth of *Streptomyces hygroscopicus* [66] is the first naturally occurring spiro-hydantoin-ribofuranose with strong herbicidal and plant growth activities toward annual, biennial, and perennial weeds by action as an adenylo-succinate synthetase inhibitor without showing toxicity to microorganisms and animals (LD_0_ > 1000 mg/kg to mammals). Several synthetic methods affording hydantocidin have been described [63,106,107,108,109,110,111,112] including application of the Bucherer–Bergs reaction starting from suitably 2,3,5-tri-*O*-protected D-ribofuranose (see entries A, B, and C in Scheme 22) [113]. However, this multistep-reaction procedure affords, like most of the other available synthetic methods, only low overall yields of hydantocidin and, therefore, is not suitable for large-scale preparation utilized for practical purposes.

## 5. Comparison with Other Methods Affording Hydantoins

The importance of hydantoins in the synthesis of biologically active compounds has led and is still leading to the development of many methods for their preparation. Although various attractive synthetic methods are available and some of them take advantage of the Bucherer–Bergs reaction, their applicability can differ significantly depending on the starting building blocks and the required substitution or functionalities on the final products. In this respect, the availability of starting reaction components as well as reaction outcomes and the ease by which the Bucherer–Bergs reaction is executed distinguish this approach from related methods leading to the formation of 5-substituted or 5,5-disubstituted N-1 and N-3 unsubstituted hydantoins.

In addition to the discussed Bucherer–Bergs reaction and its Hoyer modifications, the most important synthetic methods suitable to generate hydantoins are (a) the Read-type reaction of amino acids (or nitriles) with inorganic isocyanates; (b) the condensation of ureas with carbonyl compounds (including the Beller method for monocarbonyl compounds and the Biltz synthesis for α-dicarbonyl compounds); (c) reactions of α-amino esters with amines and phosgene and, by analogy, reactions of α-amino acid amides with ethyl chloroformate to produce urethans, followed by aqueous or alcoholic alkali-mediated cyclization; (d) the reaction of malonamides with hypohalite; (e) multi-component Ugi/De-Boc/Cyclization methodology; and (f) the modified Bucherer–Bergs reaction. Many other sophisticated syntheses of hydantoins were described (like conversion of some three-, five- or six-membered heterocycles to hydantoins, conversion from purines, solid-phase organic syntheses, combinatorial syntheses, cycloaddition reactions, cycloelimination release strategies including acid- or base-catalyzed cyclizations and thermal cycloeliminations, separate cyclization, and cleavage steps strategies; these and several others are summarized in a recent review [72]) but because of their high specificities and substantial differences in starting (or further reacting—for several-step reactions) compounds, the comparison with the Bucherer–Bergs reaction would be quite difficult or even impossible. Therefore, this section covers only comparisons with the first five mentioned methods, which are more related to the Bucherer–Bergs reaction.

The Read reaction (the reaction of free α-amino acids with sodium cyanate under acidic conditions [11,35,114,115], frequently known under the alternative name of the Urech hydantoin synthesis [116], is used for this reaction when potassium cyanate is employed) (Scheme 23) or its modifications (the reaction of α-amino nitriles with inorganic cyanate or organic isocyanate; the reaction of α-amino acids or esters with isocyanates via the intermediate ureido acids; the two-step procedure when free α-amino acids is treated with potassium cyanate in pyridine followed by acid cyclization [117]) are very good alternative spirohydantoin ring construction methods.

Because of variations in preparing the intermediary α-amino nitrile in the first step, this reaction worked even in such cases where the classical Bucherer–Bergs reaction failed. For example, attempts to form spirohydantoin from starting 1,2:5,6-di-*O*-isopropylidene-α-D-glucofuranos-3-ulose using the conditions of the Bucherer–Bergs reaction (KCN, (NH_4_)_2_CO_3_, MeOH–H_2_O, 75 °C) were unsuccessful, and the corresponding cyanohydrin was obtained as the exclusive product. In this case, the glyco-α-amino nitrile was prepared in high yield by the modified Strecker reaction using titanium(IV) isopropoxide as a mild Lewis acid catalyst and TMSCN as a cyanide source. This glyco-α-amino nitrile can be successfully cyclized to spirohydantoin in the next step using a Read-type reaction or the Hoyer modification [118]. However, it is necessary to have in mind that the Read reaction, hydantoin ring synthesis via an α-amino nitrile intermediate followed by cyclization, provides kinetically controlled products, whereas thermodynamically controlled hydantoins are obtained under Bucherer–Bergs reaction conditions. On the other hand, this reaction course control might be an advantage when the kinetic products are specifically desired. For example, sorbinil can be obtained using this method in a 67% overall yield (three steps, without silica gel chromatography) [56,119] (Scheme 24) contrary to the 40% overall yield (only two steps) [90] obtained by the classical Bucherer–Bergs reaction. In this case, the catalytic enantioselective Strecker reaction of ketoimines was applied for the preparation of the intermediate amino nitrile.

Recently, a modified method combining a catalytic reaction and the Bucherer–Bergs and Hoyer’s reaction conditions has been described [120]. In this one-pot, three-step procedure, an aldehyde or ketone was reacted with liquid ammonia under catalysis of gallium(III) triflate to produce the intermediate imine. Addition of hydrogen cyanide (generated from trimethylsilyl cyanide) to this imine afforded the corresponding amino nitrile, which, upon addition of carbon dioxide and Hünig’s base (DIPEA) in the third step, provided 5-substituted or 5,5-disubstituted hydantoins (Scheme 25). Although, in some cases the yields of hydantoins are excellent and, therefore, it can be a method of choice, there are two principal inconveniences in comparison with the classical Bucherer–Bergs reaction. First, this method requires more costly starting materials, and, second, the reaction execution is more complicated because of the three-step procedure.

More flexibility as to reactants and variation of reaction conditions is valuable for the preparation of hydantoins from carbonyl compounds and ureas. Thus, the method developed by Beller [121] (reacting different aldehydes with various ureas and carbon monoxide under palladium catalysis) affords mono-, di-, and trisubstituted hydantoins (Scheme 26).

Similar advantages are provided by the Biltz synthesis introduced nearly a hundred years ago. In this respect, the base-catalyzed condensation using benzil and urea (or thiourea) is still regarded as the most straightforward synthesis of phenytoin. Several recent improvements (including application of microwave activation instead of classical heating and the use of DMSO or dioxane/H_2_O as a solvent or two-step procedure following conversion of 2-thiophenytoin to phenytoin using hydrogen peroxide) allowed the rapid synthesis of phenytoin and structurally related derivatives in higher than 80% yields (Scheme 27) [122]. Additionally, the use of a two-phase system such as aqueous KOH/*n*-BuOH and PEG 600 as a phase transfer catalyst drastically reduced the quantity of side product, increasing the yield of phenytoin (87–93%) [123].

Although the reaction of α-amino esters with amines and phosgene (or carbonyldiimidazole as a modern alternative) [16,124,125,126,127] or cyclization of urethans [128] as well as cyclization of α-ureido esters [63] are also good alternatives for the synthesis of hydantoins, these methods suffer from low availability of common intermediates—α-amino acid amides, which, in general, are prepared in several steps. However, in some cases, this synthetic approach represents the most suitable method to obtain desirable hydantoin derivatives in a reasonable yield. For example, the potent herbicide hydantocidin was synthesized using this method in a 35.2% overall yield, along with 5-*epi*-hydantocidin in a 9.6% overall yield (Scheme 28) [63].

Depending on substitution of the starting α-amino acid amide, free hydantoin or 5-substituted, 5,5-disubstituted, as well as 3,5,5-trisubstituted hydantoins can be prepared. By analogy, N-1- and N-3-unsubstituted hydantoin with a C-5 *exo*-double bond, an analogue of naturally occurring aplysinopsin, was prepared by heating a corresponding α-methylidene-α-amino ester with urea in DMF (Scheme 29) [129].

Based on the α-amino acid amide cyclization via the corresponding isocyanate intermediates generated utilizing carbonyldiimidazole or triphosgene, Nefzi and co-workers [105,130] have developed a synthetic route to the solid-phase synthesis of hydantoin and thiohydantoin compounds and libraries from resin-bound dipeptides (Scheme 30). Using different amino acids (first site of diversity—R^1^) and different alkyl groups (second site of diversity—R^2^), this method allowed preparation of a broad range of new hydantoin derivatives. Instead of triphosgene, diphosgene was also applied in solid-phase hydantoin synthesis [131].

Hydantoins can be obtained by the application of the Hofmann degradation reaction (Hofmann rearrangement) of malonamides [132,133]. In this case, the ring closure occurs via isocyanates, the indermediates involved in the reaction of amides with hypohalite (Scheme 31). Although this method, like the Bucherer–Bergs reaction, is specific for the preparation of 5-substituted and 5,5-disubstituted hydantoins, this procedure is much less convenient than the Bucherer–Bergs reaction especially because of the more difficult availability of starting malonamides.

Recently employed Ugi/De-Boc/Cyclization methodology [134] is suitable for the preparation of fully functionalized hydantoins in good yield. Aldehydes (or ketones), amines, isonitriles, methanol, and carbon dioxide act as starting materials in this five-component reaction and corresponding carbamates result as intermediates, followed by their cyclization under alkaline conditions in the next step (Scheme 32).

In a very similar and experimentally simple methodology described as a Ugi four-component condensation (U-4CC) combined with a base-induced cyclization [135], the acid component, trichloroacetic acid, acts as a carbonic acid equivalent. In this case, the synthesis of 1,3,5-trisubstituted hydantoins can be performed by a simple one-pot, two-step procedure. Although these two methods allow the facile synthesis of arrays of hydantoins with three diversity points, the preparation of 5-mono- and 5-disubstituted hydantoins unsubstituted at N-1 and N-3 is not possible and, therefore, its application, in comparison with Bucherer–Bergs reaction, is more restricted.

A recently reported [87] modified Bucherer–Bergs reaction is based on the reaction of a nitrile with an organometallic reagent such as RMgX or RLi to generate an intermediate imine, which in a subsequent reaction with KCN and (NH_4_)_2_CO_3_ affords the corresponding hydantoin (Scheme 33). This method is practical for the one-pot synthesis of 5,5-disubstituted hydantoins and the preferential selection of this strategy should be based on the following: (i) a very large number of nitriles are commercially available or readily accessible; (ii) a variety of common organometallic reagents including RMgX and RLi add to alkyl-, aryl- and heteroaryl-substituted nitriles in high yields; (iii) protonation of the intermediate metallated imine directly leads to the NH imine, an intermediate in the Bucherer–Bergs reaction.

Although the aminobarbituric acid-hydantoin rearrangement is not related to the Bucherer–Bergs reaction, this synthetic strategy described recently by Gütschow [136,137] should be mentioned because it represents an easy access to 1,5- and/or 5,5-disubstituted, 1,3,5- and/or 1,5,5-trisubstituted, and/or 1,3,5,5-tetrasubstituted hydantoins (Scheme 34 and Scheme 35).

## 6. Experimental Conditions

### 6.1. General Comments

Despite the progress that has been made in the synthesis of hydantoins, one of the most attractive aspects of the Bucherer–Bergs reaction is its experimental simplicity and reliability. A wide variety of aldehydes and ketones can be used as a relatively easily available starting material. Because of aqueous reaction conditions, there is no need for dry solvents. Most commonly, a mixture of water with ethanol (or methanol) or methanol itself is employed [138,139,140,141,142,143,144,145,146,147,148,149,150,151,152,153,154,155,156,157,158,159,160,161,162,163,164,165,166,167,168,169,170,171,172,173,174,175,176,177,178,179,180,181,182,183,184,185,186,187,188,189,190,191,192,193,194,195,196,197,198,199,200,201,202,203,204,205,206,207,208,209,210,211,212,213,214,215,216,217,218,219,220] as a solvent, and the one-step reaction products—5-substituted and 5,5-disubstituted hydantoins (unsubstituted on N-1 and N-3)—are typically formed under thermal conditions (≅50 °C to reflux) or under pressure (sealed vessel) [98,203,221,222,223,224,225,226,227,228,229,230,231,232,233,234,235,236,237,238,239,240,241,242,243,244,245,246,247,248,249,250,251,252,253,254,255]. In some cases, amides like fused acetamide, formamide, and dimethylformamide are used as a solvent [6,250,251,256]. Occasionally, the reactions are performed under ultrasonication [8,252,257,258,259,260,261,262,263,264] or under mechanochemical ball milling using a ZnO catalyst [265]. Zinc cyanide and Fe_3_O_4_-chitosan catalyst instead of KCN [266] as well as pulsed Fe electro-oxidation [267] were applied for catalytic synthesis of hydantoin derivatives. A recent review article deals with the green synthesis of hydantoins [268].

Usually, the prepared hydantoins are stable solids easily isolated and purified by simple crystallization from the suitable solvent. Chromatographical separation (if possible) is necessary only in the case when the isolation of pure enantiomers is required.

### 6.2. Note

Potassium and sodium cyanides are violent poisons. They are highly toxic by inhalation, in contact with skin, and if swallowed and must be handled using appropriate personal protective equipment. KCN and NaCN should only be handled in a fume cupboard by qualified individuals. These cyanide salts should be properly disposed of in specially designated containers. Further information can be obtained from the Material Safety Data Sheet (MSDS) available from the supplier. Ammonia gas is also very toxic by inhalation or skin contact (may be fatal if inhaled). Handling this material requires considerable caution because it is extremely harmful to the eyes. Additionally, it is corrosive and may cause serious burns.

## 7. Conclusions

Although several synthetic methods for the preparation of hydantoins have been described so far, the Bucherer–Bergs reaction represents the simplest and very effective approach, in particular to 5-substituted and 5,5-disubstituted hydantoins (unsubstituted on N-1 and N-3). Therefore, this synthetic method is still current and often used for the synthesis of biologically and pharmacologically active compounds applicable in medicine, pharmacy, or agro-industry. In this respect, the presented review covered in depth the knowledge gained during the almost century-old history of hydantoin synthesis via the Bucherer–Bergs reaction.

## Data Availability

Not applicable.

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
