# Peer review of "The Bucherer–Bergs Multicomponent Synthesis of Hydantoins—Excellence in Simplicity"

_molecules, 2021, doi:10.3390/molecules26134024_

Round 1

Reviewer 1 Report

In this review, the authors present a comprehensive summary of the synthesis of hydantoins and their conjugates via Bucherer–Bergs reaction, the scope and limitations, as well as a comparison with some other methods were highlighted. Although several reviews of the hydantoins synthesis and application have been reported, this report focused on the multicomponent synthesis, which will promote future research in this area, so the manuscript might be accepted for publication after minor revision.

  1. One related review “Lluvia Itzel López-López*, Denisse de Loera*, Ernesto Rivera-Avalos and Aidé Sáenz-Galindo, “Green Synthesis of Hydantoins and Derivatives”, Mini-Reviews in Organic Chemistry 2020; 17(2). https://doi.org/10.2174/1570193X16666181206100225” should be cited.
  2. Some chemical structures should be modified, such as the bond angles in scheme 4.
  3. In scheme 2, the reagents involved in the formation of hydantoin should be added to the reaction mechanism. For examples, how the carbonyl transfer to hydroxy and amino group should be clarified.
  4. One C-N bond of compound 33 in scheme 9 seems incorrect.
  5. The yields of the products in all of the schemes should be marked, and all of the compounds should be marked with numbers.
  6. In scheme 23, the compound was afforded in 100% yield and 98% ee, but the authors didn’t clarify the configuration of the chiral center.

Author Response

We accepted almost all suggestions:

  1. Related review is now cited (reference 271).
  2. The bond angles in Scheme 4 were modified.
  3. Scheme 2 - the reagents were added to the reaction mechanism and the carbonyl transfer is now clear.
  4. Scheme 9 - C–N bond in compound 33 is OK.
  5. We believe that for those Schemes where individual compounds are not discussed in the text, the numbering of compounds and the yields of products are not necessary. In addition, these schemes are mostly of general importance and in most cases, the compounds are designated by a trade name.
  6. Scheme 23 (now Scheme 24) - the configuration of the chiral center is now clarified.

Reviewer 2 Report

This review is a complete and thorough examination of Bucherer-Bergs reaction. In the past some reviews have been published on the various synthetic protocols leading to hydantoins, but a review aimed exclusively at this reaction was lacking in the panorama of the chemical literature.

Authors provide a comprehensive scenario examining the reaction both from the mechanistic and applicative point of view. However, I am not convinced by the approach of chapter 5, in which the authors intend to compare Bucherer-Bergs with alternative syntheses of hydantoin. First of all, I would suggest moving chapter 6 before chapter 5 which would thus become the last chapter. Then, in my opinion, sub-chapters should be made, one for each alternative synthetic protocol proposed, indicating what are the disadvantages (or advantages) compared to the Berger reaction

Finally, a general consideration, which is beyond the quality of this review. It is very clear that there are few references relating to the last five years. This is most likely due to the fact that the use of cyanides and relatively harsh  conditions make this reaction not so "modern"

I therefore wonder if this review, on an old reaction with little possibility of restyling, could be of interest to a large audience of readers.

Author Response

According to our opinion, the inclusion of chapter "Experimental" after chapter "Comparison with other methods" seems more logical because there are no Schemes and relevant discussion in this chapter (Experimental).

As for the mention of "old reaction with little possibility of restyling", we would like to point out that the Bucherer–Bergs reaction is still one of the most widely used methods for the preparation of hydantoins. There are some possibilities to improve reaction conditions (for example search for milder reaction conditions, seeking to replace KCN with another source of cyanide ions, etc.).

Reviewer 3 Report

Please check the language particularly concerning minor mistakes and some spell issues; some sentences (underlined in yellow) should be re-phrased; check my (few) notes in the draft; 

Author Response

Most of the reviewer's comments were accepted and the text (in pdf file) was corrected accordingly:

- line 56: because there is no good evidence to suggest that Phenytoin is a human carcinogen, the text  (recently, Phenytoin is suggested to be a human carcinogen)  was deleted.

- line 188: we believe that the phrase   is tolerant of   is correct.

- line 232: because compound 38 was not indicated in the schemes, we inserted a new Scheme 11 and all subsequent schemes (and compounds) were renumbered.

- Scheme 18 (now Scheme 19): Yes, it is sorbinil (and the formula is correct).